# Influence of the Cholinergic System on the Pathogenesis of Glioblastoma: Impact of the Neutrophil Granulocytes

**DOI:** 10.3390/ijms27010321

**Published:** 2025-12-27

**Authors:** Alejandra Infante Cruz, Paula María Saibene Vélez, Cynthia Arasanz, Micaela Rosato, Federico Remes Lenicov, Juan Iturrizaga, Martín Abelleyro, Marianela Candolfi, Eleonora Regueira, Gladys Hermida, Mónica Vermeulen, Silvia Berner, Francisco José Barrantes, Silvia de la Vega, Carolina Jancic, Marcela Solange Villaverde, Gabriela Verónica Salamone

**Affiliations:** 1Instituto de Medicina Experimental (IMEX–CONICET)–Academia Nacional de Medicina, Buenos Aires C1425AUM, Argentinacjancic@gmail.com (C.J.); 2Departamento de Química Biológica, Facultad de Ciencias Exactas y Naturales, Universidad de Buenos Aires, Buenos Aires C1053ABH, Argentina; 3Centro de Educación Médica e Investigaciones Clínicas “Norberto Quirno” (CEMIC), Instituto Universitario, Buenos Aires C1037ACA, Argentina; 4Instituto de Investigaciones Biomédicas en Retrovirus y SIDA (INBIRS), Universidad de Buenos Aires–CONICET, Buenos Aires C1053ABH, Argentina; 5División Neurocirugía, Instituto de Investigaciones Médicas A Lanari, Universidad de Buenos Aires, Buenos Aires C1053ABH, Argentina; 6Instituto de Investigaciones Biomédicas (INBIOMED UBA-CONICET), Facultad de Medicina, Universidad de Buenos Aires, Buenos Aires C1053ABH, Argentina; 7Laboratorio de Biología de Anfibios-Histología Animal, Departamento de Biodiversidad y Biología Experimental, Facultad de Ciencias Exactas y Naturales, Universidad de Buenos Aires, Buenos Aires C1053ABH, Argentina; 8Departamento de Microbiología, Parasitología e Inmunología, Facultad de Medicina, Universidad de Buenos Aires, Buenos Aires C1053ABH, Argentina; 9Servicio de Neurocirugía de la Clínica y Maternidad Santa Isabel, Buenos Aires C1406DKG, Argentina; 10Molecular Neurobiology Division, BIOMED UCA-CONICET, Buenos Aires C1107AAZ, Argentina; rtfjb1@gmail.com; 11Sección Anatomía Patológica del Hospital de Quemados “Arturo U. Illia” CABA, Buenos Aires C1424BSD, Argentina; 12Unidad de Transferencia Genética y Laboratorio de Metabolismo, Departamento de Biología, Facultad de Medicina, Instituto de Oncología Ángel H. Roffo, Área Investigación, Universidad de Buenos Aires, Buenos Aires C1053ABH, Argentina

**Keywords:** acetylcholine, glioblastoma, neutrophils, M3 muscarinic acetylcholine receptor

## Abstract

Glioblastoma (GBM) is the most common malignant primary brain tumor in adults. Since numerous studies highlight the significance of cholinergic system components in tumor development, acetylcholine (ACh) and the differential activation of its receptors could play a crucial role in GBM progression. The aim of this study was to test this hypothesis by assessing the relevance of the cholinergic system in GBM cells and their microenvironment. We analyzed bulk RNA-seq expression data using the TIMER2.0 web server, focusing on the impact of patient survival in relation to muscarinic receptors (CHRM) and neutrophil infiltration in low-grade glioma (LGG) and GBM. Our analysis revealed a marked decrease in survival associated with all CHRMs, particularly in LGG. Moreover, GBM showed higher neutrophil infiltration and reduced survival, especially in relation to CHRM3. These findings were validated in the U251 cell line and in human GBM tumor biopsies (GBM-b), which also displayed CHRM3 expression. Additionally, we show that GBM cells exposed to cholinergic stimulation exhibited increased vascular endothelial growth factor (VEGF), IL-8 production, and PD-L1 expression, while the VEGF increase was blocked by tiotropium (Tio), a CHRM3 antagonist. Similarly, polymorphonuclear cells from GBM patients (PMN-p) displayed increased PD-L1 expression and IL-8 production upon cholinergic stimulation. Finally, as we previously reported on the relevance of thymic stromal lymphopoietin (TSLP) in GBM pathophysiology, here, we found that TSLP upregulated CHRM3 expression. Our findings highlight the importance of the cholinergic system in the tumor microenvironment, where it may act directly on tumor cells or influence neutrophil physiology, thereby modulating tumor progression.

## 1. Introduction

A diffuse glioma, glioblastoma (GBM) is the most aggressive type of malignant cerebral tumor [1] and represents the most common malignant primary brain tumor in adults. GBM patients have a median survival of approximately 15 months after diagnosis [2]. Standard treatment—which includes surgery, radiotherapy, and temozolomide (TMZ) therapy—is unable to fully eradicate the tumor, mainly due to its highly infiltrative nature. The range of available therapies is therefore currently limited [3]. 

Brain tumors are characterized by strong activation and infiltration of immune cells [4,5,6]. Malignant gliomas such as GBM are highly immunosuppressive, thereby favoring tumor progression through immune cell–induced microenvironmental changes [7]. Neutrophils largely infiltrate necrotic areas, likely playing an important role in glioma physiopathology. These infiltrating cells may either represent a response to necrotic lesions or actively promote tumor cell death and destruction [8]⁠. Neutrophils can adopt diverse phenotypes within the tumor microenvironment and have been classified under different terminologies, including N1/N2 neutrophils, tumor-associated neutrophils (TANs), and polymorphonuclear myeloid-derived suppressor cells (PMN-MDSCs) [9,10,11]. Fridlender et al. first proposed a functional N1/N2 polarization model, in which IFN-β drives antitumor N1 features and TGF-β skews cells toward protumor N2 characteristics [11,12,13]. However, this binary classification has evolved into a broader view acknowledging the phenotypic and functional heterogeneity of TANs, including subsets defined by density gradients such as high-density (HDN) and low-density neutrophils (LDN) [11,14]. Recent scRNA-seq studies further support a continuum of activation states rather than discrete N1/N2 categories, identifying multiple transcriptionally distinct TAN subsets associated with inflammation, angiogenesis, antigen presentation, and tumor progression [15].

Both circulating neutrophils and TANs in cancer patients retain some functional plasticity and may undergo “alternative activation” in response to the tumor microenvironment [14,16]⁠. TANs may be part of tumor-promoting inflammation processes by driving angiogenesis, extracellular matrix remodeling, metastasis, and immunosuppression. Conversely, neutrophils can also mediate antitumor responses by directly killing tumor cells and participating in cellular networks that mediate antitumor immunity. The diversity and plasticity of neutrophils underlie the dual potential of TANs in the tumor microenvironment [17,18,19]. 

Most studies on the role of neutrophils in brain tumors have focused on the impact of these infiltrating cells on antiangiogenic therapy and vascularization, a hallmark of high-grade gliomas. In this scenario, neutrophils appear to contribute to antiangiogenic therapy resistance [20]⁠. Moreover, increased neutrophil infiltration correlates with acquired resistance and progression to higher-grade gliomas in advanced stages [8,21]⁠. Preclinical studies further suggest that neutrophils are also associated with an increase in the glioma stem cell larger niche, contributing to glioblastoma progression, a process dependent on S100 proteins [21]⁠. Similarly, neutrophil activation, measured by CD11b upregulation, has been described as an early predictor of GBM progression [22]⁠.

Acetylcholine (ACh) is the neurotransmitter of both parasympathetic ganglionic and post-ganglionic nerves, as well as a non-neuronal paracrine mediator produced by various cell types [23,24]. ACh is synthesized from acetyl coenzyme A and choline in a reaction catalyzed by choline acetyltransferase (ChAT). The biological effects of ACh are mediated through two receptor families: nicotinic (CHRN) and muscarinic (CHRM) receptors. These receptors are widely distributed across tissues and can modulate various functions depending on the specific receptor subtype that is activated [25]. CHRM are G protein-coupled receptors comprising five subtypes (M1–M5), encoded by CHRM1–CHRM5 [25]. CHRM expression varies across tissues. The widely distributed expression of CHRM in different human organs suggests additional roles in other biological processes in addition to synaptic transmission [26,27,28,29]. Beyond their classical physiological roles, CHRMs are also expressed in immune cells, where they influence T cell activation and differentiation; activated CD4^+^ and CD8^+^ T cells upregulate several CHRMs subtypes, and expression has also been reported in B cells, macrophages, and dendritic cells [30,31,32,33]. Lastly, ACh is rapidly degraded by acetylcholinesterase (AChE).

In recent years, the role of ACh has been shown to extend beyond parasympathetic neurotransmission, appearing to play an important part in the local regulation of numerous cellular functions in a wide variety of non-neuronal systems, including the immune system [27,32].

Increasing evidence indicates that cancer-related cell processes such as proliferation, apoptosis, angiogenesis and even epithelial–mesenchymal transition (EMT) are mediated by overexpression of CHRN in different kinds of tumors. In breast cancer, CHRNA7 and CHRNA9 were reported to be oncogenic [34]. Research on the role of CHRM in breast cancer tumorigenesis is mainly restricted to the M3-subtype of CHRM. Since CHRM and CHRN are expressed in both central and peripheral nervous systems and in non-neuronal tissues, there is an urgent need for the development of subtype-specific CHR antagonists capable of inhibiting cancer progression without disrupting the physiological functions of the cholinergic system [35].

Numerous studies highlight the relevance of cholinergic system components in the development of neoplasms [36,37,38]. It is therefore not unreasonable to consider that ACh and the differential activation of its receptors could play a crucial role in GBM development, although this remains largely unexplored. Guizzetti and coworkers [39] demonstrated that ACh induces glial cell proliferation via CHRM—primary M3—while Ferretti et al. [40] showed that CHRM2 agonists inhibit GBM cell proliferation. Furthermore, recent research revealed that glioma patients with overexpression of CHRNA9 or CHRNA10 exhibit significantly shorter overall survival [41]. Moreover, some stress signals, such as the activation of growth factors such as platelet-derived growth factor receptor (PDGF-R), epidermal growth factor receptor (EGFR), or the activation of various protein kinase C (PKC) isoforms—key regulators of proliferation and invasion in GBM—can induce the expression of AChE [42], specifically the AChE-R splicing variant. It has been consistently established that AChE-R is directly involved in GBM proliferation [43]. AChE mRNA accumulates in primary human astrocytomas, and its expression correlates directly with tumor aggressiveness [44,45]. Interestingly, whereas AChE levels are very low in normal glial cells, they are elevated in astrocytoma tumors [46].

As previously mentioned, one of the key stress signaling pathways is mediated via the EGFR. Among the various genetic alterations identified in GBM, those affecting EGFR play a pivotal role. EGFR alterations are present in approximately 60% of GBM cases, frequently involving gene amplification and the expression of truncated receptor variants resulting from genomic deletions.

Recent studies highlight that immune cells are an important non-neuronal source of neurotransmitters enabling bidirectional crosstalk between the nervous and immune systems. When activated, neutrophils release ACh and catecholamines that can cause feedback onto the neuronal network source and transfer the neuronal signal to other immune cells, including neutrophils [47]. Indeed, peripheral human granulocytes are a non-neural source that synthesize and store ACh, but they do not produce significant amounts of ACh in comparison to lymphocytes. ChAT has been detected in human skin neutrophils, peripheral blood neutrophils [48,49] and peripheral granulocytes [27,50], M3, M4, and M5; M1 or M2 subtypes of CHRM have not been detected in neutrophils by immunocytochemistry [51] or RT-PCR [52].

Neutrophils also have all the required enzymatic machinery for the synthesis, metabolism, storage, and uptake of catecholamines. Different authors confirm that human, rat, and murine neutrophils produce catecholamines through a mechanism similar to that reported in neurons [47]. Recently, it has been reported that neutrophil extracellular trap (NET) formation could be induced by engaging CHRM3 in neutrophils [53].

In summary, at least in part, due to the complex interaction of GBM cells with the brain microenvironment and their tendency to aggressively infiltrate normal brain tissue, GBM frequently invades supratentorial brain regions that are richly innervated by neurotransmitter projections, most notably cholinergic in nature [54]. The aim of this study was to investigate the role of endogenous ACh in GBM, with particular emphasis on its capacity to modulate the tumor microenvironment through neutrophil granulocytes.

## 2. Results

### 2.1. Expression of Muscarinic Receptors in Glioblastoma

Accumulating evidence has revealed that tumor cell biology events such as proliferation, apoptosis, angiogenesis and even EMT involve CHRM overexpression in different types of tumors [35]. Here, the influence of CHRMs and neutrophils on GBM patient survival was evaluated by RNA-seq expression analysis. Figure 1A shows neutrophil infiltration and samples from GBM patients with higher CHRM3 expression (*p* = 0.029), indicating significantly lower patient survival. Association between overall CHRM expression and higher neutrophil infiltration in low grade glioma (LGG) also showed reduced patient survival (Appendix A).

These results suggest a direct association between overall CHRM expression and disease progression, as well as neutrophil infiltration in LGG. CHRM3 expression appears to be most relevant in GBM patients.

We next focused on evaluation by immunocytochemistry of the expression of CHRM1 and CHRM3, the two muscarinic receptor subtypes expressed in GBM, although CHRM1 does not appear to be relevant for patient survival. First, we assessed the expression of these two CHRMs in U251 and U373 cell lines (Figure 1B). CHRM3 expression was confirmed by flow cytometry (Figure 1C).

Subsequently, GBM biopsies (GBM-b) were analyzed by immunochemistry. Interestingly, the GBM-b showed CHRM expression in all biopsy samples evaluated (Figure 1D–H and Appendix A). Finally, we confirmed CHRM1 and CHRM3 expression in biopsies after two months of culture (GBM-bc) (Figure 1I,J). 

### 2.2. Presence of Choline Acetyltransferase and Acetylcholinesterase in Glioblastoma

Beyond neurons, different cell types play a crucial role in communication between the nervous and immune systems. Such cell types include glial cells (microglia, astrocytes, and oligodendrocytes), endothelial cells, and immune cells such as macrophages, which contribute to the release of neurotransmitters and neuropeptides, enabling complex interactions between the two systems. Based on these premises, we next evaluated whether GBM produced or degraded ACh. To this end, we assessed the presence of ChAT in the U251 and U373 cell lines by immunocytochemistry. Strong ChAT expression was observed and subsequently confirmed in U251 cells by flow cytometry (Figure 2A,B). In contrast, AChE expression evaluated by RT-PCR was poorly detected in the U251 cell line and in the GBM-b samples, irrespective of stimulation with EGF or TSLP (Figure 2C,D). Figure 2E shows higher AChE expression in PMN cells from GBM patients (PMN-p), regardless of treatment.

Bulk RNA-seq expression analysis was performed next with the TIMER2.0 web server to assess the impact of ChAT and AChE expression on patient survival in GBM, with or without considering neutrophil infiltration. No significant differences in ChAT expression were observed in either LGG or GBM. The association between ChAT and AChE expression, neutrophil infiltration, and patient survival in LGG is significant only in the context of high neutrophil infiltration, regardless of whether ChAT and AChE expression levels are high or low. In contrast, no such association is observed in GBM (Figure 2F).

### 2.3. Relevance of the Cholinergic System in 3D Glioblastoma Cultures

We decided to analyze the relevance of the cholinergic system in multicellular spheroids, an in vitro system that represents a more physiologically relevant model than monolayers by better emulating 3D architecture, nutrient gradients, and drug responses. For this purpose, U251 spheroids were cultured with or without a cholinergic agonist, either ACh or carbamylcholine chloride (Carb), and images of the spheroids were captured after 3, 6 and 9 days by phase-contrast microscopy. After 6 and 9 days of treatment with ACh or Carb, spheroids showed an increase in both diameter and volume, without impacting on the number of satellites (small spheroids) surrounding the central spheroid (Figure 3A–D). Despite a mild upward trend, we did not find a significant increase in the proliferation, either in 3D cultures (measured by acid phosphatase activity) or 2D (measured by MTS) (Figure 3E and 3F, respectively). 

### 2.4. Evaluation of Apoptosis in Glioblastoma Cells in the Presence of Cholinergic Agonists

Given that ACh has been characterized as an essential growth and survival factor with anti-apoptotic properties [55], we investigated its role in GBM spheroid apoptosis. To explore this, U251 spheroids were treated with TMZ in the presence or absence of a cholinergic agonist. After overnight culture, apoptosis was evaluated by flow cytometry using annexin-V staining. No differences were observed between spheroids cultured with TMZ in the presence or absence of cholinergic agonist (Figure 4A–C, Appendix A). Similar results were obtained using the 2D model (Figure 4D). 

### 2.5. Cholinergic Agonists Increased IL-8 and VEGF Production and PD-L1 Expression in the U251 Cell Line 

Interleukin (IL)-8 is a cytokine induced by ACh in many cell types [56] and is one of the most important chemoattractants for neutrophils. We therefore studied IL-8 secretion in U251 cells stimulated or not with ACh or Carb. After overnight treatment, IL-8 production was quantified by ELISA. Figure 5A shows that IL-8 production increased in U251 cells upon stimulation with ACh or Carb in a 2D model.

An exploratory analysis of bulk RNA-seq expression preformed with the TIMER2.0 web server presented a positive correlation between CHRM3 and IL-8 in both LGG (*p* = 0.0001) and GBM (*p* = 0.045), with no effects on CHRM1, ChAT or AChE (Figure 5B,C). The PD-1 pathway maintains immunological homeostasis and protects against autoimmunity. Programmed death-ligand 1 (PD-L1) expression on GBM cell surface promotes programmed death 1 (PD-1) receptor activation in microglia, leading to the negative regulation of T cell responses [57]. To evaluate the relevance of ACh and Carb in the pathophysiology of the tumoral microenvironment, PD-L1 expression was assessed in U251 cells cultured with or without ACh or Carb in both 2D and 3D models for 18 h. Significant increases in PD-L1 expression were observed in 2D cultures treated with ACh and Carb compared to controls, while in 3D cultures this effect was only observed with Carb (Figure 5D–G).

The exploratory analysis of RNA-seq by TIMER2.0 showed a positive correlation between CHRM3 and PD-L1 in both LGG (*p* < 0.0001) and GBM (*p* = 0.02), whereas no correlation was observed between CHRM1, ChAT, or AChE, showing in all cases a wider dispersion of the analyzed data (Figure 5H,I).

### 2.6. Muscarinic Acetylcholine Receptor M3 Mediated an Increase in VEGF Production in the U251 Cell Line

High vascular endothelial growth factor (VEGF) levels are commonly observed in GBM [58], leading us to evaluate whether stimulation with cholinergic agonist could be relevant in the production of this angiogenic factor. Our experimental findings were consistent with what we found in the exploratory RNA-seq expression analysis using the TIMER2.0 web server, showing a statistically significant positive correlation between CHRM3 and VEGF in GBM (*p* = 0.0003). The analysis between ChAT and VEGF in GBM also shows a positive correlation (*p* < 0.0001) (Figure 6A,B).

After 18 h of treatment, VEGF production was determined by ELISA in conditioned medium on U251 cells stimulated with or without ACh or Carb, and with or without the specific CHRM3 antagonist Tiotropium (Tio). In U251 3D (Figure 6C,D) and 2D cultures (Figure 6H,I), VEGF production significantly increased after stimulation with ACh or Carb. Interestingly, VEGF production was inhibited when cells were pre-incubated with Tio prior to ACh or Carb treatment (Figure 6E,F,J,K). No significant differences were found when U251 cells were stimulated with acetylcholinesterase inhibitor rivastigmine (Riv, 5 mM) (Figure 6G,L). Surprisingly, ICAM-1 expression was increased upon stimulation with ACh or Carb, and this effect was blocked by Tio (Figure 6M).

### 2.7. IL-8 and VEGF Production and PD-L1 Expression by Neutrophils from Healthy Donors or Glioblastoma Patients

To explore the relevance of the cholinergic system in the pathophysiology of neutrophils within the tumor microenvironment, we evaluated PD-L1 expression [59] in PMN cells from healthy donors (PMN-h) cultured with or without U251 cells and in the presence or absence of ACh or Carb for 18 h. Flow cytometry revealed that Carb significantly increased PD-L1 expression in PMN-h co-cultured with the U251 cell line (Figure 7A–C). PD-L1 expression was also evaluated in PMN-p (Figure 7D,E) and PMN-h (Figure 7J,K) treated or not with ACh for 18 h. No significant differences were observed in PD-L1 expression in PMN-h treated with or without ACh, whereas PMN-p showed significantly increased PD-L1 expression following ACh treatment.

To further evaluate PMN function, CD11b expression and IL-8 production were studied. PMN-h or PMN-p was stimulated or not with ACh for 15 min at 37 °C, and CD11b expression was subsequently analyzed by flow cytometry. In parallel, IL-8 secretion by PMN-h or PMN-p stimulated or not with ACh after overnight culture was also evaluated by ELISA.

CD11b expression levels and IL-8 production were found to be similarly increased in PMN-p (Figure 7F–H) and PMN-h (Figure 7L–N) when stimulated with ACh. In conclusion, ACh induces the activation of both PMN-p and PMN-h without impairing the function of PMN-p, which remains intact.

Previous studies have described higher resistance to anti-angiogenic treatment and poorer responses to chemotherapy in patients with increased TAN [21]. We therefore speculated that peripheral neutrophils might produce VEGF and that ACh could be involved in the production of this angiogenic factor. After 18 h of PMN culture in the presence of ACh, VEGF production was assessed by ELISA. As shown in Figure 7I,O, no differences in VEGF production were observed in either PMN-p or PMN-h cultured with the cholinergic agonist.

### 2.8. Thymic Stromal Lymphopoietin Increased Muscarinic Acetylcholine Receptor M3 Expression in the U251 Line 

As previously reported by our group, TSLP plays a relevant role in the pathogenesis of GBMs with neutrophil infiltration in their microenvironment [60]. To assess whether TSLP could modulate CHRM3 expression, U251 cells were cultured in the presence or absence of TSLP. Following overnight incubation, CHRM3 expression was evaluated by immunocytochemistry. As shown in Figure 8A,B, CHRM3 expression significantly increased in cells treated with TSLP for 18 h.

## 3. Discussion

GBM is the most aggressive malignant cerebral tumor [1]. Current treatments for glioblastomas include TMZ combined with radiotherapy [61], and more recently, the use of antiangiogenic agents [62,63]. Tumors actively modulate the immune response by producing factors that attract immune cells and subsequently alter their ability to recognize and respond effectively against tumor cells [64]. In this work, we analyzed RNA-seq expression using the TIMER2.0 web server, focusing on the impact of patient survival in relation to CHRM expression and neutrophil infiltration in LGG and GBM. These analyses revealed a significant decrease in survival associated with the expression of all CHRM subtypes studied. In addition, higher neutrophil infiltration correlated with reduced survival in GBM, particularly in cases with high CHRM3 expression and high neutrophil infiltration. The expression of CHRM3 was confirmed in the two cell lines studied, as well as in GBM-b and GBM-cb. Furthermore, we showed that GBM cells in the presence of cholinergic agonists displayed increased VEGF and IL-8 production, as well as enhanced PD-L1 expression. Analogously, PMN-p showed an increase in PD-L1 expression and IL-8 production when the PMN were stimulated with the cholinergic agonists. 

Our results are consistent with those of Thompson and coworkers [54] regarding the relevance of the CHRM3 in GBM. Using RNA-Seq data from The Cancer Genome Atlas (TCGA), these authors demonstrated that human GBM cell lines and patient-derived xenograft lines expressed CHRM. Here, we confirmed the expression of M1 and M3 CHRM subtypes by RNA-seq. In addition, we demonstrated that neutrophil infiltration is a relevant factor in GBM pathophysiology. Moreover, we experimentally showed the expression of M1 and M3 CHRM in GBM-b and detected the complete cholinergic system, including AChE and ChAT, in GBM cell lines. Thompson and coworkers also showed that CHRM3 activation did not alter cell proliferation or migration, although it significantly increased cell invasion through complex extracellular matrices mediated by enhanced activity of matrix metalloproteinase-9 (MMP-9) in GBM cells. In line with their findings, we did not observe differences in cell proliferation.

As previously mentioned, angiogenesis is one of the most evident hallmarks of most tumors, clearly distinguishing GBM from normal brain tissue [58,65]. Hence, anti-angiogenesis therapy could constitute an effective strategy for treating GBM patients. It has been shown that VEGF plays an essential role in angiogenesis in GBM; thus, inhibiting VEGF expression may be an effective therapeutic strategy for combating GBM growth in neurological practice [66,67]. However, antiangiogenesis induced by the monoclonal antibody anti-VEGF, bevacizumab, has shown minimal efficacy and furthermore it enhances tumor invasiveness through hypoxia induction [68,69]. Neutrophil infiltration contributes to antiangiogenic therapy resistance [20],⁠ and a positive relationship has been reported between increased neutrophil infiltration in tumor tissue, acquired resistance, and higher-grade glioma at later stages [8,21].

Different studies have demonstrated the relevance of antagonists of CHRM3 as a possible treatment in tumoral proliferation [70,71]; in particular, studies in gastric cancer cells have shown increasing concentrations of CHRM3 antagonists like 4-DAMP and darifenacin in a model of tumor xenograft growth in nude mice [72]. The role of CHRM in tumor angiogenesis has still not been fully characterized. Lombardi et al. demonstrated that carbachol increased the constitutive expression of VEGF-A in tumor cells, an effect that was reverted by the muscarinic antagonist atropine [73]. In addition, in BALB/c mice bearing LMM3 mammary adenocarcinoma cells, administration of the cholinergic agonist Carb, with or without various muscarinic antagonists, increased VEGF expression [74,75]. Furthermore, tumor macrophages stimulated angiogenesis via activation of CHRM1 and CHRM2, which triggered the arginine metabolic pathway [74]. In experiments focused on VEGF production, Anna Folino et al. [76] demonstrated that tiotropium suppressed the ACh-induced increase in VEGF in both control and asthmatic fibroblasts. Our results show participation of the cholinergic system in VEGF production in the U251 line cell. This production was blocked by TIO, thus confirming the predominant participation of CHRM3 in this phenomenon

Moreover, as previously described, the adhesion molecule ICAM-1 is an important surface glycoprotein involved in tumor invasion and angiogenesis in gliomas [77]. Here, we observed that ICAM-1 was modulated by ACh and Carb and was inhibited completely by Tio, further confirming the role of CHRM3 in this cholinergic-mediated mechanism. 

Increasing evidence in recent years has shown that CHRM3 may play a vital role in the carcinogenesis of many types of cancer [78], including colon cancer [79], lung cancer [80], and cholangiocarcinoma [81]. Notably, CHRM3 was overexpressed in human gastric cancer tissue, and correlated with cancer stage and lymph node metastasis [82]. Huangfei Yu et al. [72]. suggested that CHRM3 plays a major role in ACh-stimulated gastric cell proliferation. They demonstrated that ACh could act through CHRM3 to activate the EGFR pathway, while CHRM3 knockdown or EGFR inhibition could reverse the cell proliferation and phosphorylation of ERK and AKT induced by ACh stimulation in gastric cancer by activating non-canonical signals [72] (non-canonical signals include Ras–Raf-1–Erk–AKT for the CHRM3 receptor subtype [83,84]. The data suggest that EGFR is an indispensable molecule in the process of ACh-stimulated gastric cancer cell proliferation; however, the exact mechanism of action between CHRM3 and EGFR is still unclear in gastric cancer. Studies in colon cancer revealed that CHRM3 was able to facilitate ACh-induced EGFR signaling activation [85].

Under our experimental conditions, ACh did not significantly increase 3D proliferation but enhanced spheroid volume, suggesting that ACh may promote tumor cell fitness and spheroid expansion through non-proliferative mechanisms, such as cell volume regulation and changes in spheroid organization

In our system, the increase in spheroid volume may include cellular osmotic or fluid accumulation, modulation of ion and calcium fluxes, and/or changes in spheroid compaction. Acetylcholine is known to regulate intracellular calcium signaling and ion transport, processes closely linked to cell volume control and tissue dynamics [86].

Consistent with this interpretation, multicellular spheroid expansion has been shown to occur through mechano-osmotic mechanisms without changes in proliferation [87].

Another relevant observation of the present work is the significant increase in IL-8 production when either PMN-h or PMN-p was cultured with the cholinergic agonist ACh. It is well established that IL-8 is a chemokine that promotes neutrophil recruitment by increasing PMN both in the tumor microenvironment and in the periphery, two phenomena associated with poor prognosis GBM patients [21]. In addition, IL-8 secretion has been reported to establish a pro-tumorigenic microenvironment and facilitate cancer progression and metastatic spread through autocrine and paracrine pathways [88]. Here, we report increased IL-8 production in the U251 cell line when cultured with the cholinergic ligands ACh or Carb. Expression of CHRM3 conditions GBM patients to a worse prognosis by enhancing neutrophil recruitment and promoting the protumorigenic effects of IL-8.

GBM cells induce PD-L1 expression by activating various receptors such as Toll-like receptor (TLR) and EGFR [57]. Relevant studies have demonstrated that PD-L1 expression in gliomas correlates with World Health Organization (WHO) grading and could thus be considered a tumor biomarker [57]. Interestingly, we observed that PD-L1 increased in the U251 cell line, in PMN-p, and in PMN-h co-cultured with U251 cells in the presence of ACh. Sun and colleagues [59] demonstrated increased PD-L1 expression in TAN. In the present work, our findings highlight the relevance of ACh in inducing an increase in PD-L1-mediated neutrophil infiltration in GBM.

The PD-L1/PD-1 axis may also play a role in regulating VEGF levels that mediate angiogenesis [89]. In 64 patients with primary glioma Xue et al. demonstrated that PD-L1 expression levels correlate well with VEGF levels [90]. However, not all studies agree on this point [53,90]. Research by Joseph and coworkers indicate a negative relationship between VEGF-related genes and PD-L1/PD1 axis activity [89]. Our results suggest that ACh may represent a common factor between VEGF production and PD-L1 expression, at least in the U251 cell line.

The analysis performed using the web server TIMER2.0 [91,92] also revealed a positive correlation between CHRM3 expression and the expression of IL-8, PD-L1 and VEGF, confirming our experimental findings indicating that the expression of CHRM3 and PD-L1 and the production of IL-8 and VEGF tend to increase together in glioblastoma tumors.

Our group previously demonstrated the presence of TSLP in GBM tumor cells and its possible role as a modulator of neutrophil physiology in the tumor microenvironment [60]. We observed that GBM tumor cells of U251 or GBM-b express TSLP when stimulated with EGF. Moreover, the combination of TSLP with neutrophil infiltration is associated with worse patient prognosis and survival, suggesting that GBM may contribute to immunomodulation of the tumor microenvironment through the production of TSLP, as previously reported [93,94,95]. In this work, we also demonstrate that TSLP increased CHRM3 expression in the U251 cell line, enhancing tumor malignancy, probably by further promoting the complex interaction of GBM cells with the brain microenvironment and their tendency to aggressively infiltrate normal brain tissue.

Our work highlights the relevance of the cholinergic system in the tumor microenvironment, mainly through CHRM3. These receptors may act directly on the tumor or influence neutrophil physiology within the tumor microenvironment. The clinical and therapeutic significance of this paper is that it suggests that Tio could serve as an inhibitor of VEGF production in GBM. This blockade could contribute to glioblastoma treatment, potentially leading to a better prognosis or improved quality of life for patients.

## 4. Materials and Methods

### 4.1. Reagents and Antibodies

Ficoll–Hypaque was from GE Healthcare Bio-Sciences AB (Uppsala, Sweden). RPMI 1640 culture medium and fetal bovine serum (FBS) were from Invitrogen (Carlsbad, CA, USA). Dulbecco’s Modified Eagle’s Medium (DMEM; 12800017) was from ThermoFisher (Carlsbad, CA, USA). PE-conjugated mouse antibody directed against CD11b and the corresponding isotype controls were from BD Biosciences (San Jose, CA, USA). ACh and Carb were from Sigma-Aldrich (St. Louis, MO, USA). Anti-PD-L1 and anti-EGFR antibodies were from BioLegend (San Diego, CA, USA). TMZ was obtained from Sigma (St. Louis, MO, USA). TSLP was from BioLegend (San Diego, CA, USA). In some experiments, cells were pre-incubated with a selective CHRM3 antagonist, Tio 30 nM (Boehringer Ingelheim, Buenos Aires, Argentina). CHRM expression was evaluated using specific goat IgG polyclonal antibodies directed against M1 or M3 subtypes (Santa Cruz Biotech, Heidelberg, Germany), followed by secondary FITC-labelled polyclonal IgG antibodies against goat primary antibodies (Sigma-Aldrich, St. Louis, MO, USA).

### 4.2. U251 Cell Culture

The U251 glioma cell line was cultured in DMEM supplemented with 10% FBS [96]. Cells were cultured at a concentration of 5 × 10^4^ cells/100 µL in 96-well flat-bottom plates and grown until confluence. U251 cells were then co-cultured with neutrophils. The same protocol was applied to the U373 cell line.

### 4.3. U251 Spheroids 

U251 cells were seeded on top of 1.5% solidified agar to form spheroids (5 × 10^5^ cells/mL) and incubated in medium containing varying concentrations of ACh and Carb. After 9 days of cholinergic agonist treatment, spheroid size (area and volume) was determined by microscope, and cell viability was quantified using the acid phosphatase (APH) assay [97].

### 4.4. Histology and Immunochemistry

GBM biopsy tissue was fixed with 4% formaldehyde, dehydrated and embedded in paraffin:histoplast (50:50, *w*/*w*). Serial transverse sections were performed, and the samples were subsequently deparaffinized, rehydrated, and used for histology and immunochemistry. 

For antigen recovery, sections were boiled in 10 mM L21 sodium citrate (pH 8) for 10 min. To block non-specific staining, sections were incubated for 1 h with blocking buffer containing 5% Bovine serum albumin (BSA) in PBS. Next, primary antibody was applied overnight to the sections in a humid chamber. Anti-AChT antibody (mAb5350), was obtained from Chemicon International (Oakville, ON, Canada). Goat anti-CHRM1 (C-20, sc-7470) and goat anti-M3 antibody CHRM (H-20, sc-31486) were purchased from Santa Cruz Biotechnology (Heidelberg, Germany). Secondary FITC-labelled polyclonal IgG antibodies (Alexa Fluor 488) directed against goat IgG were from Sigma-Aldrich and applied for 45 min.

### 4.5. Fluorescence Microscopy

Samples were imaged using an FV1000 confocal microscope (Olympus, Tokyo, Japan) confocal microscope or a Zeiss model 980 confocal microscope (Oberkochen, Germany).

### 4.6. Biopsy Disaggregation 

GBM-b (Table 1) were incubated with collagenase (2 mg/mL) and DNAse (1000 IU) for 30 min at 37 °C, then inactivated with 10% FBS and 2 mM EDTA.

Cells were incubated at 5 × 10^6^ cells/mL in medium containing ACh (10^−8^ M, 10^−9^ M) and Carb (10^−8^ M, 10^−9^ M) at different concentrations overnight at 37 °C. Subsequently, cells were washed with PBS and used for RT-PCR or fixed with 1% paraformaldehyde (PFA) for flow cytometry analysis.

### 4.7. Biopsy Mechanical Disaggregation

Tumor samples were obtained following surgical resection from a GBM patient who had provided written informed consent. Surgical procedures were carried out either at the División de Neurocirugía del Instituto de Investigaciones Médicas Dr. Alfredo Lanari or at the División de Neurocirugía de la Clínica y Maternidad Santa Isabel, both in Buenos Aires, Argentina.

Primary cell lines derived from surgically resected biopsies were established by mechanical dissociation of tumor fragments in serum-free culture medium. Cells were subsequently cultured in DMEM supplemented with 10% FBS and penicillin (100 U/mL) and streptomycin (100 μg/mL) at 37 °C in a humidified atmosphere containing 5% CO_2_. Serial passages were performed by trypsinization. The cells were washed with PBS 1x and incubated for 2 min with trypsin 1% at 37 °C, after which the trypsin activity was inactivated with FBS 10%.

GBM-bc [41] cells were characterized by optical microscopy and confirmed to be free of fibroblast contamination.

### 4.8. Blood Samples

Blood samples were obtained from 6 healthy donors and from 9 GBM patients (Table 1) who had not taken medication for at least 10 days prior to sampling. Blood was collected by venipuncture of the forearm vein directly into heparinized plastic tubes.

### 4.9. Neutrophil Isolation

Neutrophils were isolated from peripheral blood using Ficoll–Hypaque gradient centrifugation and dextran sedimentation (Ficoll, GE Healthcare, Munich, Germany; Dextran, Alfa Aesar, Tewksbury, MA, USA) as previously described [98]. Cells were resuspended in RPMI-1640 supplemented with 10% FBS, or 1% FBS for apoptosis assays only. The resulting preparations were evaluated by flow cytometry, which confirmed a neutrophil purity >98% [60] (since most PMN are neutrophils, the terms will be used interchangeably hereafter). Only samples with <0.5% monocytes were included in the assays.

### 4.10. Cell Cultures

Neutrophils (90 μL, 5 × 10^6^ cells/mL) were seeded onto flat-bottom 96-well plates in the presence or absence of varying concentrations of ACh or Carb (10^−8^ M–10^−9^ M). After overnight incubation at 37 °C and 5% CO_2_, supernatants and cells were recovered. Cytokine production was measured using ELISA and apoptosis was evaluated either by fluorescent microscopy or flow cytometry.

### 4.11. Neutrophils and Glioblastoma Co-Culture Assay

Neutrophils (90 μL, 5 × 10^6^ cells/mL) were resuspended in RPMI supplemented with 10% FBS and co-cultured with the U251 cell line in the presence or absence of ACh or Carb (10^−8^ M) for 18 h at 37 °C. As a control, neutrophils were cultured alone in the presence or absence of ACh or Carb (10^−8^ M–10^−9^ M). After overnight incubation, supernatants were recovered for ELISA and the CD11b expression assay. All the co-culture experiments were performed in RPMI medium supplemented with 10% FBS.

### 4.12. Cell-Surface CD11b Expression

Neutrophils (90 μL, 5 × 10^6^ cells/mL) were cultured with or without ACh or Carb (10^−8^ M–10^−9^ M) for 15 min at 37 °C and labeled with a PE-conjugated anti-CD11b monoclonal mouse antibody in 0.5% BSA and 2 mM EDTA in PBS for 15 min at 4 °C. Cells were then washed with PBS and fixed with 1% PFA for flow cytometry analysis.

### 4.13. Expression of Muscarinic Receptors in Biopsy

A total of 1 × 10^6^ U251 cells were fixed in 2% formaldehyde in PBS for 15 min at 4 °C. Cells were washed twice with PBS, resuspended in cold methanol for 5 min at 4 °C and washed again. Cells were incubated with glycine/PBS (0.1 mg/mL). Conventional immunostaining was performed [32]. CHRM3 and ChAT were evaluated using specific goat IgG polyclonal antibodies directed to the M3-subtype CHRM (Santa Cruz Biotech, Germany), followed by secondary FITC-labeled polyclonal IgG antibodies directed to goat or rabbit IgG (Sigma-Aldrich). Analysis was performed using FACSCalibur and FACSCanto flow cytometers and CellQuest (BD Biosciences, San Jose, CA, USA) and FlowJo v10 (BD Biosciences, San Jose, CA, USA).

### 4.14. Expression of Muscarinic Receptors in U251

A total of 1 × 10^6^ U251 cells were cultured in 24-well plates containing pre-sterilized glass coverslips previously treated with poly-L-lysine (0.01 mg/mL) to promote cell adhesion. After 24 h of culture, the attached cells were fixed with cold 3% PFA for 20 min at room temperature. Samples were then washed three times with PBS.

Subsequently, cells were incubated for 1 h in PBS containing 0.1 M glycine, followed by additional PBS washes. Next, a single wash was performed with PBS containing 0.05% saponin and 0.5% BSA for 20 min on ice, after which the solution was removed.

Primary antibodies were then added: anti-CHRM1 (1:50) and anti-CHRM3 (1:100), and incubated overnight at 4 °C in a humid chamber. The following day, PBS washes were performed and the secondary antibody (goat polyclonal IgG conjugated to FITC, Sigma-Aldrich) was applied at a 1:100 dilution for 45 min. Finally, four gentle washes were carried out with PBS–saponin–BSA, followed by a final wash with PBS. For mounting, coverslips were air-dried and 10 µL of Aquapolymount were applied.

### 4.15. Cytokine Production Assay

IL-8 and VEGF production by the cells was quantified by ELISA kits according to the manufacturer’s protocol. Human IL-8 ELISA kit was obtained from R&D Systems (Minneapolis, MN, USA), and VEGF ELISA kit was obtained from Biolegend (San Diego, CA, USA).

### 4.16. Quantitation of Apoptosis by Annexin-V Binding and Flow Cytometry

Annexin-V binding to U251 cells was performed using an apoptosis detection kit according to the manufacturer’s instructions (Biolegend, San Diego, CA, USA) and analyzed by flow cytometry (FACScan flow cytometer; Becton Dickinson, San Jose, CA, USA). Results are expressed as a percentage of annexin-V-positive cells.

### 4.17. AChE Expression by RT-qPCR

RT-qPCR (reverse transcription followed by quantitative polymerase chain reaction) was performed to evaluate AChE expression in the U251 cell line and in GBM-b, in PMN-h and PMN-p, and in the GBM-b or U251 cells. Total RNA was extracted using TRIzol^®^ (Invitrogen) according to the manufacturer’s instructions. RNA was reverse-transcribed into cDNA using random primers and M-MLV reverse transcriptase (Invitrogen). Using specific primer pairs for GAPDH as the housekeeping gene and AChE at 10 µM each, cDNA amplification was performed in real time with SYBR^®^ Green PCR Master Mix (Applied Biosystems, Foster, CA, USA) on a Real-Time StepOne Plus thermocycler. All reactions were performed in duplicate. The expression level of the AChE target gene was quantified using the ΔΔCt method and expressed relative to GAPDH.

Primer sequences:

AChE Forward: CGAAACTACACGGCAGAGGA

AChE Reverse: CGCAGGTCCAGACTAACGTA

Actin Forward: AGGCATCCTCACCCTGAAGT

Actin Reverse: GCGTACAGGGATAGCACAGC

### 4.18. Samples and Datasets

153 GBM RNA-seq samples from the Illumina HiSeq 2000 platform were obtained from The Cancer Genome Atlas (TCGA) (URL: https://www.cancer.gov). Survival curves for low and high expression of CHRM, and correlation analysis of expression levels between IL-8, VEGFA, and CD274 were performed using the TIMER2.0 web server (URL: http://timer.cistrome.org) [91,92].

### 4.19. Statistical Analysis

Statistical analyses were performed using GraphPad Prism 8.4.2 (La Jolla, CA, USA). Statistical significance for multiple comparisons was determined using the nonparametric Kruskal–Wallis test with Dunn’s post-test, and Friedman test for multiple comparisons with Dunn’s post-test. For comparison between two groups, nonparametric Mann–Whitney U tests (unpaired samples) or Wilcoxon tests (paired samples) were used. Significance was defined at *p* < 0.05 (*) or *p* < 0.01 (**). Experiments were partially blinded. Immune association analyses between gene expression groups were performed using TIMER2.0, with significance defined at *p* < 0.05.

## Figures and Tables

**Figure 1 ijms-27-00321-f001:**
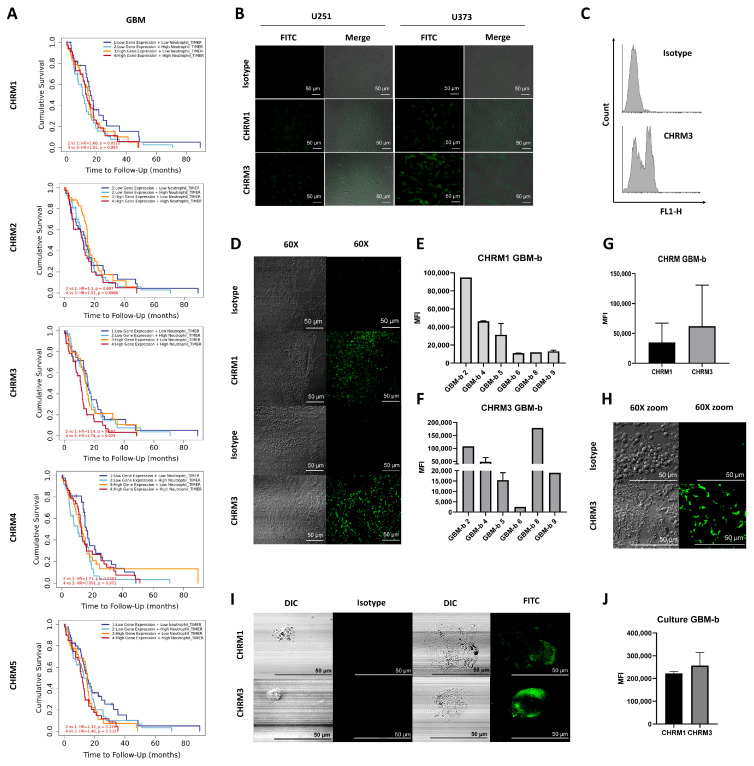
Relevance of CHRM3 expression in GBM patients. (**A**) Meta-analysis of TCGA patient gene expression databases evaluating CHRM1, CHRM2, CHRM3, CHRM4, and CHRM5 expression in GBM samples and their correlation with neutrophil infiltrate and overall survival. Analysis performed with TIMER2.0 (Tumor Immune Estimation Resource; http://timer.cistrome.org/) GBM = 153. Spearman correlation, *p* < 0.05. (**B**) Expression of cholinergic system components in GBM cell lines (U251 and U373). CHRM1 and CHRM3 expression evaluated by immunostaining using a Nikon Eclipse Ti-E fluorescence microscope (200×). (**C**) CHRM3 expression in U251 cells assessed by flow cytometry. Representative experiment of three performed. (**D**) CHRM 1 and CHRM3 expression in GBM-b samples evaluated by immunostaining (60× magnification). Representative experiment from six patients. (**E**,**F**) Quantification of mean fluorescence intensity (MFI) at 20×. Mean ± SD. (**G**) Comparison between CHRM1 and CHRM3 expression (MFI) across six GBM-b patient samples. Mean ± SD. (**H**) CHRM3 60× zoom. (**I**) CHRM1 and CHRM3 expression in GBM-bc cells after two months of culture, assessed by immunostaining (60× magnification). Representative experiment. (**J**) Quantification of MFI of GBM-bc after two months of culture. Mean ± SD.

**Figure 2 ijms-27-00321-f002:**
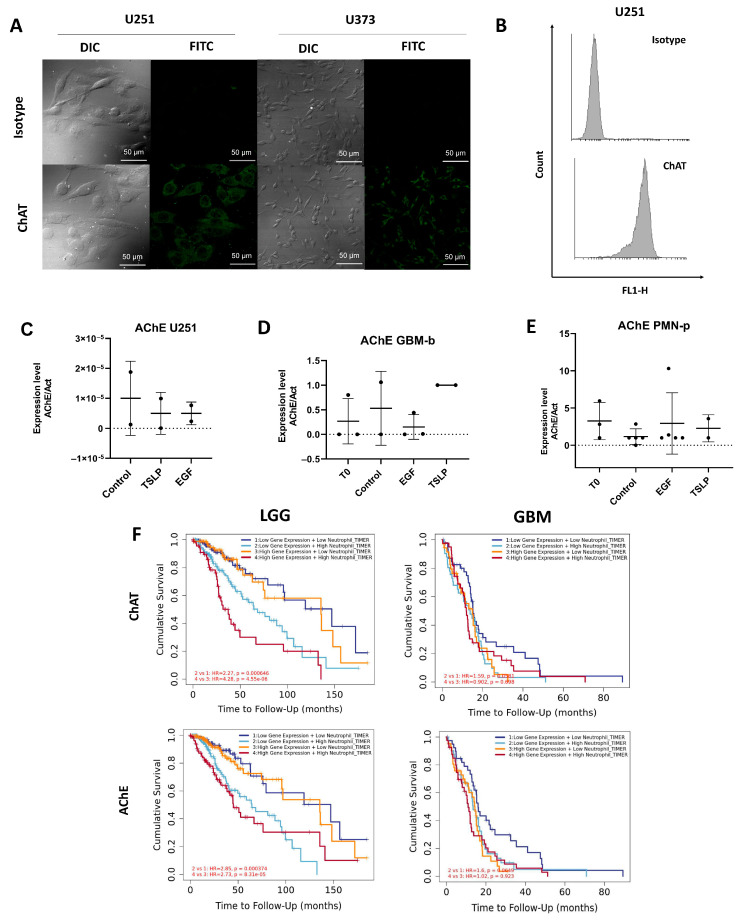
AChE and ChAT expression in GBM. (**A**) ChAT expression in U251 and U373 cell lines analyzed by immunostaining and confocal microscopy (60× magnification, Nikon Eclipse Ti-E fluorescence microscope). (**B**) ChAT expression in U251 cells assessed by flow cytometry. (**C**) AChE expression in U251 cells, (**D**) in primary cells from GBM-b (3 × 10^6^ cells/mL) and (**E**) in PMN-p (3 × 10^6^ cells/mL) under basal conditions (T0) or after stimulation with EGF (10 ng/mL) and TSLP (25 ng/mL). Cells were cultured for 24 h at 37 °C, harvested, and analyzed by RT-PCR. (**C**–**E**) figures show AChE expression relative to actin. (**F**) ChAT (top) and AChE (**bottom**) expression in LGG (**left**) and GBM (**right**) samples obtained from the TCGA database.

**Figure 3 ijms-27-00321-f003:**
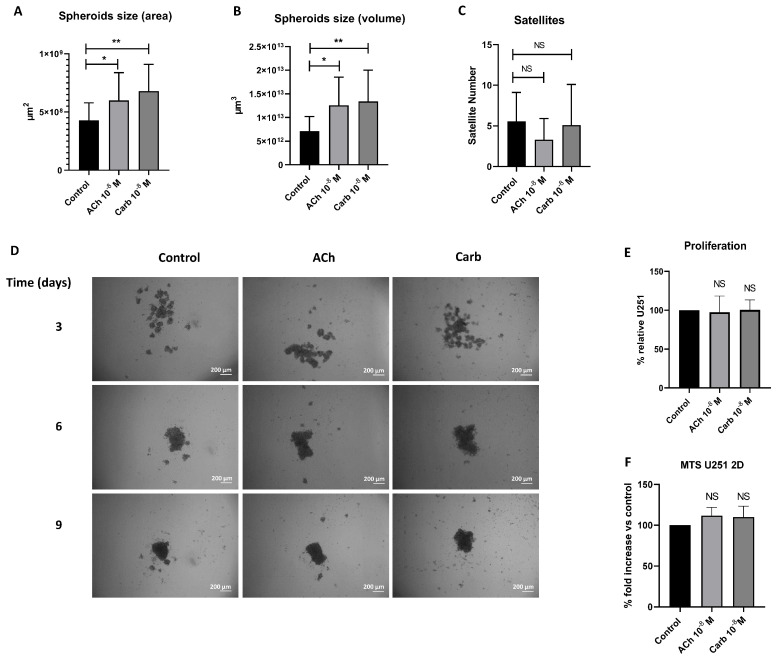
The cholinergic system promotes morphological changes in U251 spheroids. U251 cells were seeded on 1.5% solidified agar to form spheroids (5 × 10^4^ cells/mL) and incubated with medium containing ACh or Carb. Phase-contrast microscope images were obtained with a Nikon Eclipse™ E400 fluorescence microscope and captured using a Nikon Coolpix R995 digital camera at 40× magnification. (**A**) Quantification of spheroid area, (**B**) volume, and (**C**) number of satellites on day 9. Results are expressed as mean ± SD; non-parametric unpaired Mann–Whitney U test: U251 + ACh day 9 vs. U251, (* *p* = 0.0383, n = 28); U251 + Carb vs. U251 (** *p* = 0.0031, n = 27). (**D**) Representative photographs taken at different time points. (**E**) Percentage of U251 spheroid proliferation measured as acid phosphatase activity and (**F**) MTT assay after 9 days of culture relative to control.

**Figure 4 ijms-27-00321-f004:**
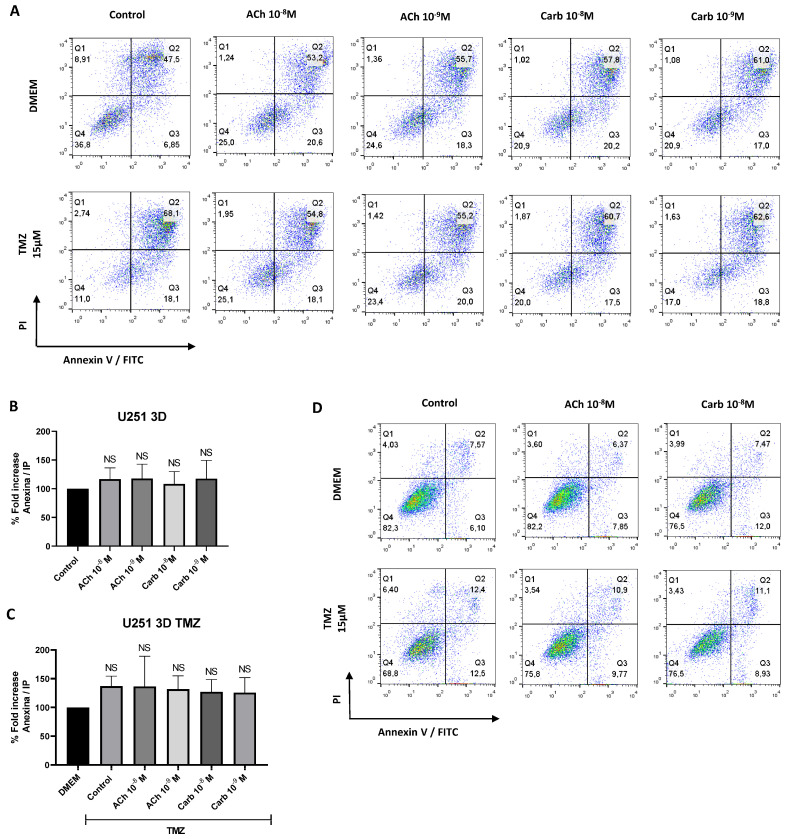
Evaluation of apoptosis in GBM cells in the presence of cholinergic agonists. (Top panel **A**,**B**) U251 cells were cultured for 9 days in the presence or absence of ACh (10^−8^–10^−9^ M) or Carb (10^−8^–10^−9^ M). (Bottom panel **A**,**C**) After 8 days of culture with the mentioned treatments, cells were treated with 15 µM TMZ for 24 h. Cell viability was assessed after 24 h using the Annexin V/Propidium Iodide kit by flow cytometry. (**A**) Representative dot plot of Annexin V vs. PI in cells treated with ACh or Carb. (**B**) Percentage increase in Annexin V^+^ cells after ACh or Carb treatment, and (**C**) after TMZ treatment, both relative to control. (**D**) Apoptosis evaluation in U251 cells in 2D culture exposed to ACh (10^−8^–10^−9^ M) or Carb (10^−8^–10^−9^ M). Results are expressed as mean ± SD; n = 5.

**Figure 5 ijms-27-00321-f005:**
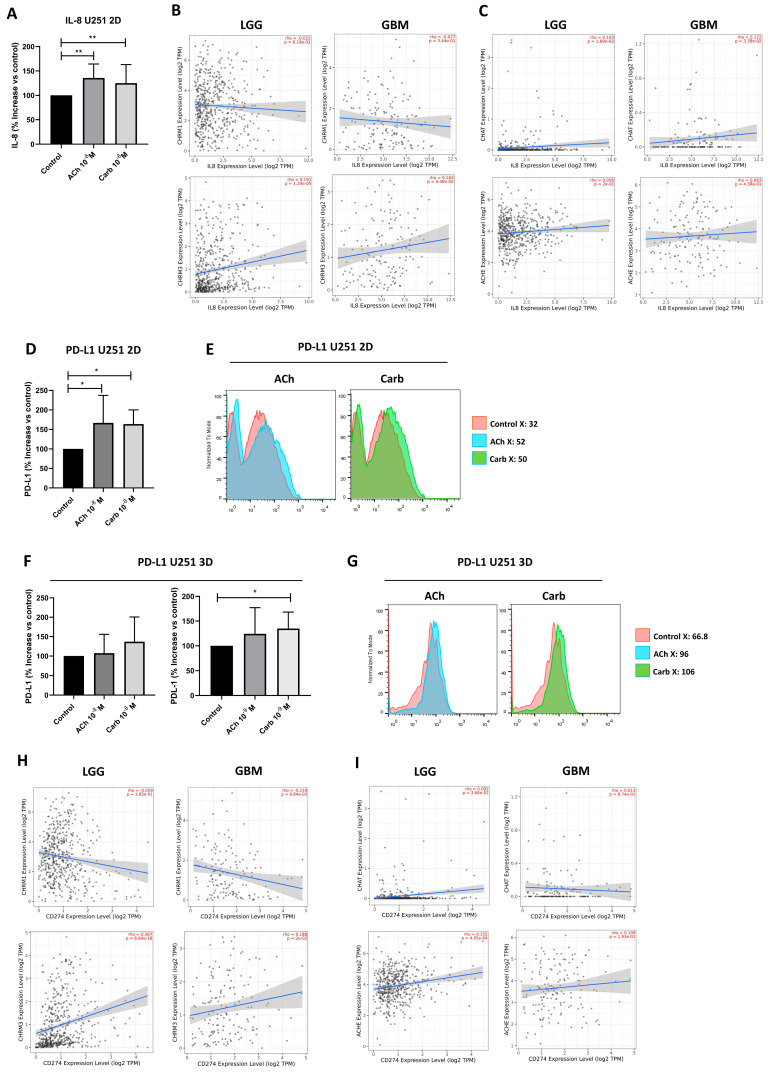
Influence of the cholinergic system on IL-8 production by U251 tumor cells. (**A**) IL-8 levels in culture supernatants measured by ELISA after culture. Ten experiments were performed in the 2D model (** *p* < 0.005, mean ± SD). (**B**) Correlation of CHRM1 (upper panels) and CHRM3 (lower panels) and of (**C**) ChAT (upper panels) and AChE (lower panels) with IL-8 expression in LGG (n = 516) and GBM (n = 153) samples, estimated by TIMER2.0 (http://timer.cistrome.org/). (**D**–**G**) PD-L1 expression in U251 tumor cells. (**D**) PD-L1 expression in U251 cells in 2D model (monolayer grown for 24 h with cholinergic system agonists ACh (10^−8^–10^−9^ M) or Carb (10^−8^–10^−9^ M) and (**F**) 3D model (spheroids grown for 9 days in 1.5% agar, 5 × 10^4^ cells/mL, treated with ACh (10^−8^–10^−9^ M) or Carb (10^−8^–10^−9^ M). * *p* < 0.05. (**E**) Representative experiment on 2D cultures and (**G**) 3D cultures. (**H**) Correlation of CHRM1 (upper panels) and CHRM3 (lower panels), and of (**I**) ChAT (upper panels) and AChE (lower panels), with PD-L1 expression in LG (n = 516) and GBM (n = 153) samples, estimated by TIMER2.0.

**Figure 6 ijms-27-00321-f006:**
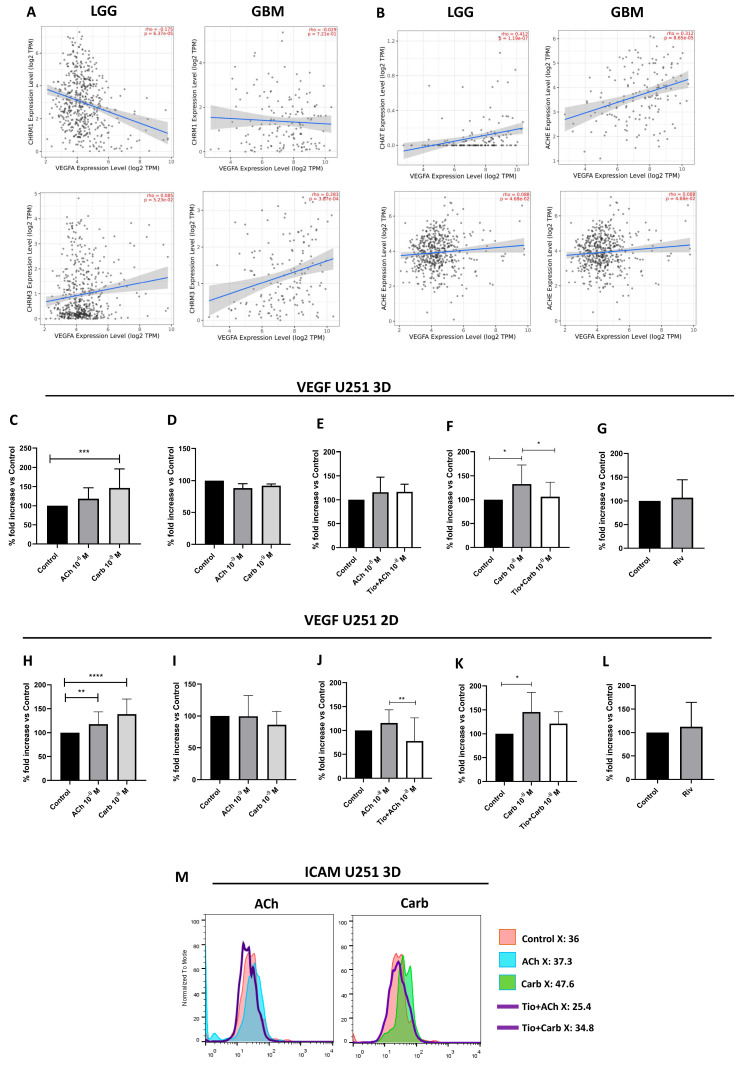
Relationship between VEGF expression and the cholinergic system in GBM. (**A**) Correlation of CHRM1 (upper panels) and CHRM3 (lower panels) with VEGF-A expression in LGG (n = 516) and GBM (n = 153) samples, estimated by TIMER2.0 (http://timer.cistrome.org/). (**B**) Correlation of ChAT (upper panels) and AChE (lower panels) with VEGF-A expression in LGG and GBM samples, estimated by TIMER2.0. (**C**–**G**) VEGF production in U251 cells cultured as 3D spheroids (9 days, 1.5% agar, 5 × 10^4^ cells/mL) and treated with ACh (10^−8^–10^−9^ M), Carb (10^−8^–10^−9^ M), Tio (30 nM, 15 min pretreatment followed by ACh or Carb 10^−8^ M), or Riv (5 μM). Supernatants were collected and VEGF levels measured by ELISA. Nine independent experiments were performed (** *p* < 0.005, mean ± SD). (**H**–**L**) VEGF production in U251 cells cultured as 2D monolayers (24 h) with the same treatments described above. Ten independent experiments were performed (* *p* < 0.05, ** *p* < 0.005, *** *p* < 0.0005, **** *p* < 0.0001, mean ± SD). (**M**) ICAM-1 expression in 3D cultures measured by flow cytometry. A representative experiment is shown.

**Figure 7 ijms-27-00321-f007:**
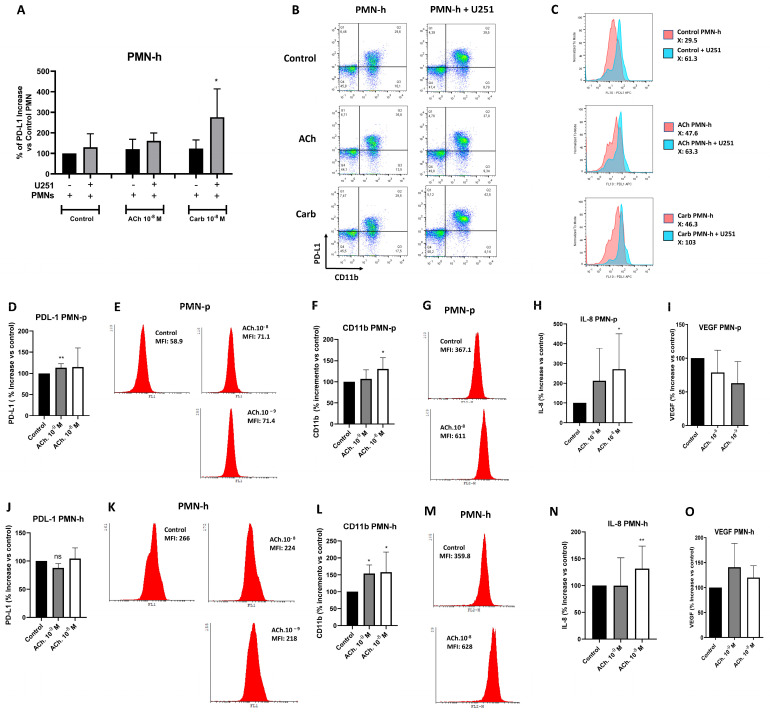
Functional evaluation of the cholinergic system in PMN-p, PMN-h and cocultures with the U251 cell line. (**A**–**C**) PD-L1 expression in PMN-h cultured (5 × 10^6^ cells/mL) with or without U251 cells for 24 h in the presence of ACh (10^−8^ M) or Carb (10^−8^ M). PD-L1 levels were measured by flow cytometry using CD11b as PMN marker. (**D**,**E**) PD-L1 expression in PMN-p cultured for 24 h with ACh (10^−8^–10^−9^ M). Seven independent experiments were performed (** *p* < 0.005). (**J**,**K**) PD-L1 expression in PMN-h cultured under the same conditions. (**E**,**K**) Representative experiment. (**F**,**G**,**L**,**M**) CD11b expression in (**F**,**G**) PMN-p and (**L**,**M**) PMN-h after 15 min stimulation with ACh (10^−8^–10^−9^ M). CD11b levels were evaluated by flow cytometry. Seven independent experiments were performed (* *p* < 0.05). (**G**,**M**) Representative experiments. (**H**,**I**,**N**,**O**) IL-8 and VEGF production in (**H**,**I**) PMN-p (**N**,**O**) and PMN-h after 24 h stimulation with ACh (10^−8^–10^−9^ M). Cytokine levels were measured in culture supernatants by ELISA. Seven independent experiments were performed (* *p* < 0.05).

**Figure 8 ijms-27-00321-f008:**
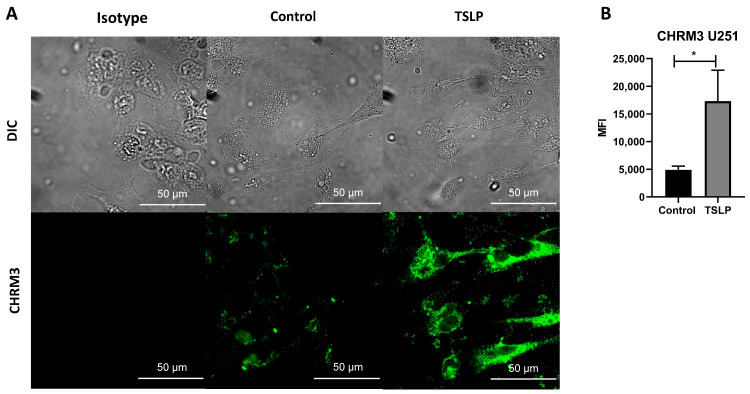
CHRM3 expression in the U251 cell line. (**A**,**B**). CHRM3 expression in U251 cells treated with TSLP. Cells (7 × 10^6^/mL) were cultured in 24-well plates and stimulated with TSLP at 70% confluence. After 24 h, CHRM3 labeling was performed and analyzed by confocal microscopy. * *p* < 0.05. Data from triplicate experiments.

**Table 1 ijms-27-00321-t001:** GBM-b (biopsy from a patient with GBM), GBM-bc (biopsy-derived cell culture), and PMN-GBM-b (polymorphonuclear leukocytes in GBM). Quantification of polymorphonuclear leukocytes was performed in histological sections of central nervous system tumors stained with H&E, counting 10 fields at 40× magnification (# = number, F = femenine, M = masculine).

Patient Code	Age	Gender	GBM-b	GBM-bc	Blood Samples	# PMN-GBM-b
GBM-b1	68	F		X	X	
GBM-b2	78	F	X	X	X	3
GBM-b3	53	F			X	
GBM-b4	45	F	X		X	7
GBM-b5	55	M	X	X	X	2
GBM-b6	58	M	X		X	6
GBM-b7	51	M	X			
GBM-b8	40	M	X	X	X	3
GBM-b9	44	F	X		X	5
GBM-b10	65	F	X		X	

## Data Availability

The data presented in this study are available in the public repository of The Cancer Genome Atlas (TCGA) (URL: https://www.cancer.gov), analyzed with the free open access web server TIMER2.0 (URL: http://timer.cistrome.org). The original contributions presented in this study are included in the article/Appendix A. Further inquiries can be directed to the corresponding author.

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
