# Peer review of "Influence of the Cholinergic System on the Pathogenesis of Glioblastoma: Impact of the Granulocyte Neutrophil"

_ijms, 2025, doi:10.3390/ijms27010321_

Round 1

Reviewer 1 Report

Comments and Suggestions for Authors

The authors uncover a pro-tumor cholinergic signaling of muscarinic receptor, CHRM3, in glioblastoma. They show that GBM cells express choline acetyltransferase and low AChE enable the accumulation of ACh within the tumor. They provide a mechanism involving CHRM3 activation that stimulates GBM cells to promote angiogenesis, neutrophil recruitment, and immune suppression. Neutrophils themselves respond to ACh with increased IL-8 and PD-L1 expression, reinforcing a positive feedback loop that amplifies inflammation and tumor progression. The cytokine TSLP enhances CHRM3 expression, linking immune cytokine signaling to cholinergic sensitivity. The proposed mechanism depicts a self-sustaining CHRM3-driven cholinergic inflammatory circuit that accelerates glioblastoma growth and immune evasion, suggesting that CHRM3 antagonists such as tiotropium could have therapeutic potential. While role of CHRM3 in cancer proliferation and invasion is known, this study shows that CHRM3 signaling links cholinergic activity to neutrophil-driven inflammation and immune suppression in GBM. This significantly adds to our understanding and is of interest.

The biggest concern is of statistical signifiance and clarity of figures. While there is mention of use of non-parametric tests in methods section, most figures lack visible p-values, significance markers, or sample sizes. In several panels, mean ± SD is shown without any statistical annotations, making it unclear which effects are significant and which are trends. The absence of these details weakens confidence in the reported associations, particularly for multi-panel figures where claims “significant increase,” “no difference” are not statistically demonstrated.

Significant improvement can be done to figure arrangement and labeling. There is a lot of gab and some figures don’t have a good resolution. Some examples are below.

Fig. 2 The bottom row in Figure 2A is ambiguous — please clarify what it represents (cell line, magnification, or staining condition). Add explicit panel labels (e.g., U251 / U373, 20× / 60×), scale bars, and consistent exposure settings for comparability.

Panels C–E would be clearer if presented using a consistent graph format. Please justify why panel C uses a bar plot while D and E are scatter plots. For consistency and transparency, consider replotting all three as individual data points with mean ± SD or SEM, and indicate replicate numbers (n = x).

Fig. 3 Provide labeled, scaled images in Panel D, showing which condition each image represents. Consistent illumination and magnification are important for fair comparison.

Fig. 4: Improve figure clarity by increasing contrast, labeling each subplot, and providing gating information or representative dot-plots in Supplementary Data. Is it statistically significant?

Author Response

We would like to thank the reviewer for his/her valuable comments and suggestions that contributed to improve the quality of the manuscript and to clarify the presentation of our findings. I answer point by point in the attachment below, and all the changes to the manuscript were highlighted.

Reviewer 2 Report

Comments and Suggestions for Authors

The manuscript explores the role of the cholinergic system, particularly CHRM3, in glioblastoma progression. The overall premise is interesting and important, and the authors provide strong experimental evidence using TCGA data, GBM cell lines, and patient-derived biopsies. However, a few areas need attention:

  1. The variability observed among GBM-b 1–6 samples for CHRM1 expression should be discussed. Is this difference biologically significant, or due to technical variation? If this reflects inter-patient heterogeneity, please mention it briefly in the results.
  2. In Figure 2A, the “isotype” and “ChAT” labels can be included directly in the figure, rather than just in the graph, for easier reading. The U251 panel appears to be taken at a higher magnification (closer to 60×) compared to U373. Please clarify and make sure the scale bars are legible and consistent.
  3. In Figure 2B, please indicate whether the differences shown are statistically significant. If not significant, consider mentioning explicitly in the text. Please mention the p-values and n-values for all the figures in the legend.
  4. The English language requires some polishing. For example, lines 408–411 can be rewritten as: “In contrast, AChE expression evaluated by RT-PCR was poorly detected in the U251 cell line and in the GBM-b samples, irrespective of stimulation with EGF or TSLP.” On line 168, “GBM frequently invade” should be corrected to “GBM frequently invades.” Similarly, in line 378, change “CHRM1 do not appear to be relevant” to “CHRM1 does not appear to be relevant.”
  5. Figures need to be of better quality. The legends, axis labels, and scale bars are not legible. While this could be a journal formatting issue, higher-resolution figures would make the results easier to evaluate during the review process.

Overall, the study presents strong data for a novel concept and is likely to be of interest to the field, but the clarity of figures, quality of presentation, and sentence construction need improvement to match the strength of the findings.

Comments on the Quality of English Language

See above 

Author Response

(The authors gave the same response as above.)

Reviewer 3 Report

Comments and Suggestions for Authors

Reference:  “Influence of the Cholinergic System on the Pathogenesis of Glioblastoma: Impact of the Granulocytes Neutrophils. Submittted to IJMS. October 2025.  Infante Cruz Alejandra  et al., 2025.  

  General comments: In the manuscript submitted for publication at IJMS, the authors are studying neuropharmacological and physiological aspects of glioblastomas, a very malignant and aggressive type of tumor. Throughout the manuscript, the authors evaluate the role of the cholinergic system (acetylcholine and its receptors) in the progression of glioblastoma. They correlated patient survival with muscarinic receptor expression levels and neutrophil infiltration in glioblastomas. The data showed a marked decrease in patient survival related to an increase in muscarinic receptors and neutrophil infiltration. The authors also showed that glioblastoma cells exposed to cholinergic stimulation exhibited increased vascular endothelial growth factor (VEGF), IL-8 production, and PD-L1 expression. They also showed that polymorphonuclear leukocytes from patients with glioblastomas showed increased PD-L1 expression and IL-8 production after cholinergic stimulation. Finally, the authors highlight the importance of the cholinergic system in the tumor microenvironment, where it can act directly on tumor cells or influence neutrophil physiology to modulate tumor progression. After reading the text, I believe this topic falls within the scope of the IJMS. The text is well-written, clear, scientifically sound, and addresses an extremely important topic for public health. Attached are suggestions and questions that will help the authors producing a revised version that is more complete and attractive to interested readers. 

Specific Comments:

1- At the end of the article summary, I missed the key words that facilitate the understanding of the text and indicate to readers the main topics of the text. 

2- In the line 71… Brain tumors are characterized by strong activation and infiltration of immune cells. Please indicate one or more references for this text!  

3-  In the lines 83 and 84, authors wrote … The diversity and plasticity of neutrophils underlie the dual potential of TANs in the tumor microenvironment (8-10). Out of curiosity for this reviewer, are there any data in the literature showing different neutrophil subpopulations, just as there are different types of lymphocytes, that justify the antagonistic effects of neutrophil infiltration in the intratumoral environment? If so, perhaps a brief introduction to these data could be included here! 

4- The text described between lines 94 to 103, which discuss the role of acetylcholine as a neuronal mediator, paracrine signaling, and its receptors, could have more specific and intermediate citations in the text, instead of just the citations at the end of the text. 

5- In the lines 101 and 102 … The widely distributed expression of CHRM in different human organs suggests additional roles in other biological processes in addition to synaptic transmission (14, 15). It is unclear whether reference 14 refer only to this last paragraph, or to the entire text between lines 91 and 103. 

6- Still about this text… The widely distributed expression of CHRM… Are there published data showing non-canonical activity for these receptors? If so, could the authors once again elaborate a bit on the text? 

7- Once again, out of curiosity for this reviewer, what is the activity of inhibitors of acetylcholine binding to these cholinergic receptors in tumor progression? If there are data in the literature showing some activity of cholinergic receptor antagonists in tumor cell biology or tumor growth, then some comment could be made in the text. 

8- The authors discuss the lack of information on the involvement of acetylcholine and its receptors in the tumor cellular biology of glioblastomas and the logic of considering this highly relevant. But what about choline, a cleavage product of acetylcholine, which has also been linked to cellular signaling? Any insights? 

9- The involvement of acetylcholinesterase, an enzyme that processes acetylcholine into choline and acetate, may be an indicator of choline's involvement in this multimolecular axis. Some comment could be made on this. 

10- The text written between lines 133 and 137 …. As previously mentioned, one of the key stresses signaling pathways is mediated via the EGFR. Among the various genetic alterations identified in GBM, those affecting EGFR play a pivotal role. EGFR alterations are present in approximately 60% of GBM cases, frequently involving gene amplification and the expression of truncated receptor variants resulting from genomic deletions.  Although relevant to the cellular tumor biology of glioblastomas, this text seems meaningless to me regarding the roles of acetylcholine, acetylcholinesterase, choline, and their receptors. Could the authors write something to tie the texts together, if such a thing exists? 

11- The text described between lines 154 and 166, where the authors discuss the role of neutrophil-secreted components in obstructive pulmonary disease, seems to me a bit distant from the tumor cellular biology of glioblastomas and could be removed from the introduction. If the authors were to discuss the entire cellular biology of neutrophils, we would have a book, or even more than one. 

12- For the text written between lines 174 and 176, the word “Abbreviations used in the text” is missing. I also suggest that the authors review this list, as many commonly used abbreviations are not present in the list. If a text with the meanings of abbreviations is shown, it should include all the abbreviations used throughout the text. 

13- I found the introduction a bit long, but well-written. The authors discuss aspects of glioblastomas, the role of neutrophils in tumor biology, and finally the role of acetylcholine and its receptors in the tumor cell biology of glioblastomas. Well-written, well-founded and relevant text for the interest and learning of readers. But see my comments on small excerpts of text that can be removed from the introduction. 

14- About M/M. Line 178 to 184. The text written between lines 178 to 184 should be preceded by title:  “Ethical Committee and experimental protocols with human”, and separated from M/M. 

15- Line 202….The U251 glioma cell line. Please indicate the origin of cells?  From whom they were purchased or obtained. 

16- Line 216, authors wrote….Biopsy tissue was fixed with 4 % formaldehyde. Please indicate what tissue(s) was or were studied! 

17- The text written between lines 215 to 227 inform about histochemistry and immunohistochemistry, but authors just detailed immunohistochemistry. How about histochemistry assays? They were performed?  

18- Lines 230 and 231 …Samples were imaged using an Olympus Fluoview 1000 confocal microscope or a Zeiss model 980 confocal microscope. Please informe detail of manufactures, contry, as informed for reagents.     

19- Line 234. ….GBM biopsies (GBM-b) were incubated. In a critical sense and scientific rigor authors must inform details of age, sex/gender and collection date of material from patient(s). 

20- Line 241…Tumor samples were obtained following surgical resection from a patient with GBM. In a critical sense and scientific rigor authors must inform details of age, sex/gender and collection date of material from patient(s).   

21- Line 250 … Serial passages were performed by trypsinization, as previously described.  Please indicate reference of previous described.

22- Line 251… After two months of GBM biopsies culture (GBM-bc) (45), and cultured cells were… I belive that the words “and cultured” can be removed!   After two months of GBM biopsies culture (GBM-bc) (45), cells were…

23- Line 255… Blood samples were obtained from healthy donors or from patients with GBM ….Please indicate the number of healthy and patients with GMB who provided samples of blood.

24- Line 260 … Neutrophils were isolated from peripheral blood using Ficoll–Hypaque. Please indicate of how many donors neutrophils were isolated?   

25- Lines 263 to 266… The resulting preparations contained >98% neutrophils (since most PMN are neutrophils, the terms will be used interchangeably hereafter). Only samples with less than 0.5 % monocytes were used for the assays. In my opinion here authors must indicate how assay they used to identify cells in the leukocyte suspension. For instance by means of a blood smear stained with methylene blue/eosin, or by means of flow cytometry with a specific antibody, or other technique. 

26- In the lines 272 and 273…. were evaluated by ELISA and by either fluorescent microscopy or flow  cytometry, respectively.  Please complement with …as detailed below.  

27- Line 276… Neutrophils were resuspended in supplemented RPMI. Supplemented with what? Fetal calf serum? Then indicate! as well as other kind of supplement.

28- Lines 276 and 277… Neutrophils were resuspended in supplemented RPMI and co-cultured with the 276 U251 cell line in the presence or absence of ACh or Carb for 18 h at 37 °C . Include details of how many cells were used. Details of concentration of ACh or Carb……  

29- The same observation for line 283. Please point details of culture and number of cells used.

30- Lines 300 and 301 … Cells were cultured in 24-well plates containing pre-sterilized glass coverslips previously treated with poly-L-lysine to promote cell adhesion. Please In a critical sense and scientific rigor authors must inform details of how many cells were used, and concentration of poly-L-lysine used.

31- Results.  Lines 363 , 400, 429 and others…This is my personal opinion, but the overuse of abbreviations not only detracts from readers' interest and diminishes the journal's prestige, but also hinders reader learning. Abbreviations should be used in long texts with repetitive definitions. In titles and subtitles, authors should write terms without abbreviations; this is more eye-catching and easier to grasp, thus enhancing the journal's prestige. Think about !!! 

32- In the line 364 …..cancer cell processes…. In my opinion I suggest authors change to Tumor cell biology events….  

33- In the lines 364 and 365 authors wrote…. Accumulating evidence has revealed that cancer cell processes such as proliferation, apoptosis, angiogenesis and even epithelial-mesenchymal transition (EMT) are mediated .  In my opinion, the word "are mediated" is too strong, giving the idea that this molecule is the main one involved in these events, and that doesn't seem to be the case. Therefore, I suggest changing it to “have the participation” . 

34- The authors need to review the presentation of the graphs shown in Figure 1A, which are important for the indicated conclusion, but even with a magnification of more than 500% it is not possible to read the captions within the figures. It's a lot of group work to be shown in low quality. Redo the figure or separate it as an individual figure with a higher magnification. 

35- Once again, Figure 1B seems too small to capture all the information. However, in this case, differences between the two cell lines can still be seen for immune labeling . 

36- Still on figure 1B, what do the authors mean by merge?  The word merge means combine or join together. But what was joined together there? The immune staining with DIC? If was DIC and immune staining what was the goal? Author must define in the legend of figure!!!  Merge is normally used when making two fluorescent markings. 

37- Figure 1C also needs to be shown at higher magnification. The meanings of the X and Y axes are not shown. I understand that on the X axis we would have fluorescence intensity in base 10 logarithmic and on the Y axis the number of events. Is that it?

38- Why are the authors showing the 20X magnification in Fig. 1D? Why not just leave it at 60X as shown in Figure 1F. The markings are much more visible and the authors could shown as zoon pointed in fig. 1F.

39- See Figure 2A. Why don't the authors show the previous 1B and 1D fluorescence at this magnification? Much more visible and elegant and without merge.

40- See also fig. 2F. At this magnification it is easy to understand the analysis performed and its results, but not at the magnification shown in 1A.

41- Any special reason why the authors show two types of graphs for similar analyses in Figures 2C, D, and E?

42-Still for figure 2, unlike analyses for ChAT, which show protein expression by fluorescence, in the case of AChE, the authors show production of transcribed mRNAs. These are different analyses. Is there any particular reason for not performing fluorescence for AChE?

43- About figure 3, lines 430 and 431, authors wrote…. multicellular spheroids, an in vitro system that closely resembles in vivo solid tumors. This is a partial interpretation, as the model also has several criticisms. Low reproducibility of spheroids, the simple nature of the cells compared to the abundance in the intratumoral microenvironment, a simpler ECM compared to real tumors, in addition to the absence of blood vessels. At some point the authors considered doing experiments with animals in vivo 

44- In my opinion, the data shown in figure 3 are very questionable, given the low reproducibility of the model and the small variations depending on the different treatments and statistical data found. 

45- About figure 3. Doesn't it seem contradictory for the authors to conclude that treatments with both ACh and Carb increased spheroid growth, but did not increase cell proliferation studied by two different methods? 

46- Why didn't the authors test the effects of the treatments done by colony forming or clonogenic assays? Or check the cell cycle using the flow cytometer that they have and propidium iodide? If treatments really increase cell proliferation, these trials would show this in addition to being more reproducible. 

47- About fig. 4. The data in Figure 4 are clear and show that the treatments did not alter phosphatidylserine expression in the cells studied. It would not be appropriate to simply show this data as "data not shown." 

48- The authors also need to remember that in the case of tumor cells, phosphatidylserine production is not necessarily related to apoptosis, but also to evasion of the immune system and resistance to chemotherapy. 

49- About fig. 5. I believe that the increased expression of IL-8 as pointed in fig. 5A, a molecule involved in neutrophil migration, could also have been evaluated by RT-q-PCR. Any reason the authors didn't do these experiments? 

50- In the case of figure 5A, the authors could show the graph with greater magnification! 

51- I'd also like to know why the authors didn't create a treatment concentration curve and instead used only just one concentration of ACh or Carb in the analysis. This would provide robust information to support the authors' hypotheses. 

52- Although the authors are confident in the data shown in Fig. 5B-D, the data seem to me to be very scattered for the parameters studied. 

53- RNA-seq analysis by TIMER2.0 has points of criticism that could interfere in results.  For instance: differences in sample preparation, and fails of algorithms.  I would like the opinion of authors about this?  

54- In my opinion, this data needs to be confirmed with more robust analyses. 

55- About results related to Fig. 6. Authors wrote beteween lines 505 and 506 ….showing a statistically significant positive correlation between CHRM3 and VEGF in GBM (p = 0.0003). Correlation does not mean causality. Therefore, these data can be interpreted with parsimony ! 

56- The same observation was pointed above is also valid to sentence …Unexpectedly, a positive correlation was also found between ChAT and VEGF in GBM (p < 0.0001) (Figures 6A-B).  

57- I don't know if I could be wrong, but in my opinion, the data shown in Fig. 6C are at the statistical limit of what the authors claim. Especially the treatment with ACh, the main agonist in this event. The authors should also repeat the experiment with time kinetics and various concentrations of the agonists tested to comporve their hypothesis! 

58- In figure 6E, in my opinion, the authors should have shown the control in the absence of treatment! 

59- Finally, it would be interesting if the authors had shown the expression levels of VEGF transcripts under the conditions tested by RT-q-PCR. Then it would be interesting to show a correlation between transcripts and protein expression. 

60- The flow cytometry data (fig. 6F) confirm the changes at the threshold of statistical significance. In my opinion, the authors should invest in more robust analyses to support this hypothesis. 

61- Once again, Figure 7A is confusing since the main agonist ACh does not alter the event in comparison with the control. A major criticism is that the authors should have performed analyses with curves of different concentrations, increasing the contractions of the agonists and checking the responses. 

62- I don't know, but once again I would like the opinions of other reviewers, but in my opinion the statistical data shown in fig. 7 do not point to degrees of confidence in the experiments done. In the figures shown, I have the impression that the authors should initially show the smallest contractions tested, i.e., 10-9 before 10-8 M, which is a higher concentration. If they are checking the influence of solute concentration, then smaller ones first, then larger ones. Well, it's a style! 

63- 

Author Response

(The authors gave the same response as above.)

Round 2

Reviewer 1 Report

Comments and Suggestions for Authors

The 

Line 247: Despite a mild upward trend, we did not find a significant increase in the proliferation...

Authors could discuss the possible biological basis for the volume increase if proliferation is unchanged. Does ACh induce cell hypertrophy or fluid accumulation? This is crucial because "tumor progression" usually implies proliferation.

Line 247: Despite a mild upward trend, we did not find a significant increase in the proliferation...

Authors could discuss the possible biological basis for the volume increase if proliferation is unchanged. Does ACh induce cell hypertrophy or fluid accumulation? This is crucial because "tumor progression" usually implies proliferation.

Typos: There could be others.

Ln 521: mechanics  -> mechanical 

Ln 301: re- gardless -> regardless

Author Response

(The authors gave the same response as above.)

Reviewer 2 Report

Comments and Suggestions for Authors

The authors have addressed all my concerns. 

Author Response

(The authors gave the same response as above.)

Reviewer 3 Report

Comments and Suggestions for Authors

Dear MDPI/IJMS friends. After carefully reading the response letter submitted by the authors, indicating changes to the manuscript, and following some of the suggestions I forwarded after analyzing the first version of this manuscript, it is my opinion that the changes performed in this revised version of the manuscript left the manuscript more complete from a scientific point of view and meet the criteria for publication by IJMS. The authors made many changes throughout the revised manuscript, following the reviewers' suggestions, making the text more attractive and complete. In the reply letter, the authors indicate what was possible to do within their technical limitations, and in my opinion, I believe that this was sufficient to make the text more scientifically complete. Therefore, it is my opinion that this revised version meets the journal's technical priorities and can be published by IJMS. If these are also the opinions of the other reviewers and the Editorial Board I think the manuscript could be published by IJMS. Greetings to the authors and congratulations on the work.

Author Response

(The authors gave the same response as above.)
